# Density-wave ordering in a unitary Fermi gas with photon-mediated interactions

Victor Helson[1,2], Timo Zwettler[1,2], Farokh Mivehvar[3], Elvia Colella[3], Kevin Roux[1,2,4], Hideki Konishi[1,2,5], Helmut Ritsch[3] & Jean-Philippe Brantut[1,2 ✉]

A density wave (DW) is a fundamental type of long-range order in quantum matter tied to self-organization into a crystalline structure. The interplay of DW order with superfluidity can lead to complex scenarios that pose a great challenge to theoretical analysis. In the past decades, tunable quantum Fermi gases have served as model systems for exploring the physics of strongly interacting fermions, including most notably magnetic ordering[1], pairing and superfluidity[2], and the crossover from a Bardeen–Cooper–Schrieffer superfluid to a Bose–Einstein condensate[3]. Here, we realize a Fermi gas featuring both strong, tunable contact interactions and photon-mediated, spatially structured long-range interactions in a transversely driven high-finesse optical cavity. Above a critical long-range interaction strength, DW order is stabilized in the system, which we identify via its superradiant light-scattering properties. We quantitatively measure the variation of the onset of DW order as the contact interaction is varied across the Bardeen–Cooper–Schrieffer superfluid and Bose–Einstein condensate crossover, in qualitative agreement with a mean-field theory. The atomic DW susceptibility varies over an order of magnitude upon tuning the strength and the sign of the long-range interactions below the self-ordering threshold, demonstrating independent and simultaneous control over the contact and long-range interactions. Therefore, our experimental setup provides a fully tunable and microscopically controllable platform for the experimental study of the interplay of superfluidity and DW order.

Quantum gas experiments provide a unique opportunity to create complex quantum many-body systems from the bottom up by starting from a dilute gas and adding interactions in a controlled way. This was initially enabled by the precise control of the intrinsic contact interaction between atoms using Feshbach resonances[4]. Recent years have seen tremendous efforts to engineer more complex many-body systems using tailored longer-range interactions[5]. As a key extension in this direction, dipolar interactions between atoms with large permanent magnetic moment were successfully used to create supersolid phases of bosons[6]. For fermions, stronger interactions promised in polar molecules[7] or transiently realized using Rydberg dressing[8] could further lead to exotic quantum phases.

Cavity quantum electrodynamics provides a flexible platform for engineering non-local, all-to-all interactions among polarizable particles mediated by cavity photons[9–11]. By loading atoms inside a high-finesse cavity and driving them with a transverse pump beam in the far-detuned, dispersive regime, an effective interaction between the atoms is produced, described by an effective interaction Hamiltonian[11],

$$\hat{H}_{\text{int}} = \iint d^3\mathbf{r}\, d^3\mathbf{r}'\, \mathcal{D}(\mathbf{r}, \mathbf{r}')\hat{n}(\mathbf{r})\hat{n}(\mathbf{r}'),\qquad(1)$$

where $\hat{n}(\mathbf{r})$ is the local density operator at position $\mathbf{r}$. In a single-mode cavity, this interaction has a spatially periodic, infinite-range structure of the form $\mathcal{D}(\mathbf{r}, \mathbf{r}') = \mathcal{D}_0 \cos(\mathbf{k}_p \cdot \mathbf{r})\cos(\mathbf{k}_c \cdot \mathbf{r})\cos(\mathbf{k}_p \cdot \mathbf{r}')\cos(\mathbf{k}_c \cdot \mathbf{r}')$, which arises from the interference of the pump and the cavity mode[12]. Here, $\mathcal{D}_0 = U_0 V_0 / \Delta_c$ is the interaction strength, with $U_0$ being the cavity potential depth per photon and $V_0$ being the light shift induced by the pump, proportional to the intensity of the pump laser. $\Delta_c$ is the detuning of the pump from the cavity resonance, whose sign determines the attractive or repulsive nature of the interaction (Methods). The wave vectors of pump and cavity photons are denoted by $\mathbf{k}_p$ and $\mathbf{k}_c$, respectively. Physically, the interaction Hamiltonian (equation (1)) describes the correlated recoils from the scattering of a pump photon off an atom into the cavity mode and back into the pump by a second atom.

This photon-mediated density–density interaction leads to the self-organization into a density-wave (DW) phase, as was first observed in thermal atoms[13], then in Bose–Einstein condensates (BECs)[14,15] and lattice Bose gases[16,17], and recently, in non-interacting Fermi gases[18]. In weakly interacting BECs, the DW self-ordering is a manifestation of the Dicke superradiant phase transition, and it allowed for the quantum simulation of supersolidity[19]. By exploiting more atomic internal levels and many cavity modes, a variety of rich phenomena ranging from magnetic ordering[20,21] to dynamic gauge fields[22] and self-ordering in elastic optical lattices[23] were observed in bosonic systems. Even more intriguing phenomena ranging from threshold-less self-ordering in low

[1]Institute of Physics, Ecole Polytechnique Fédérale de Lausanne, Lausanne, Switzerland. [2]Center for Quantum Science and Engineering, Ecole Polytechnique Fédérale de Lausanne, Lausanne, Switzerland. [3]Institut für Theoretische Physik, Universität Innsbruck, Innsbruck, Austria. [4]Institute of Science and Technology Austria, Klosterneuburg, Austria. [5]Department of Physics, Graduate School of Science, Kyoto University, Kyoto, Japan. ✉e-mail: jean-philippe.brantut@epfl.ch

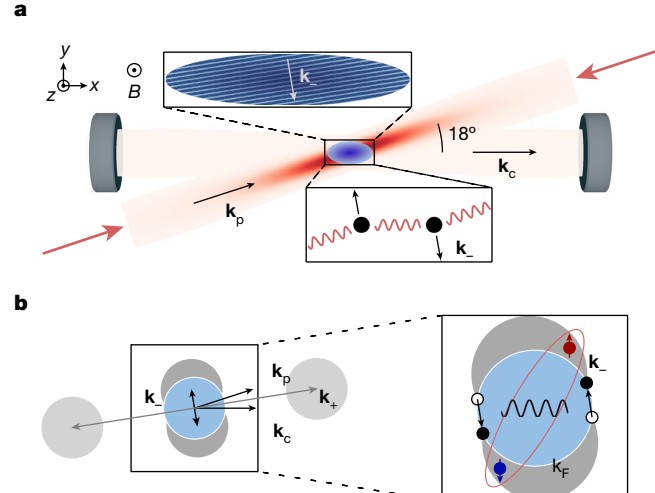

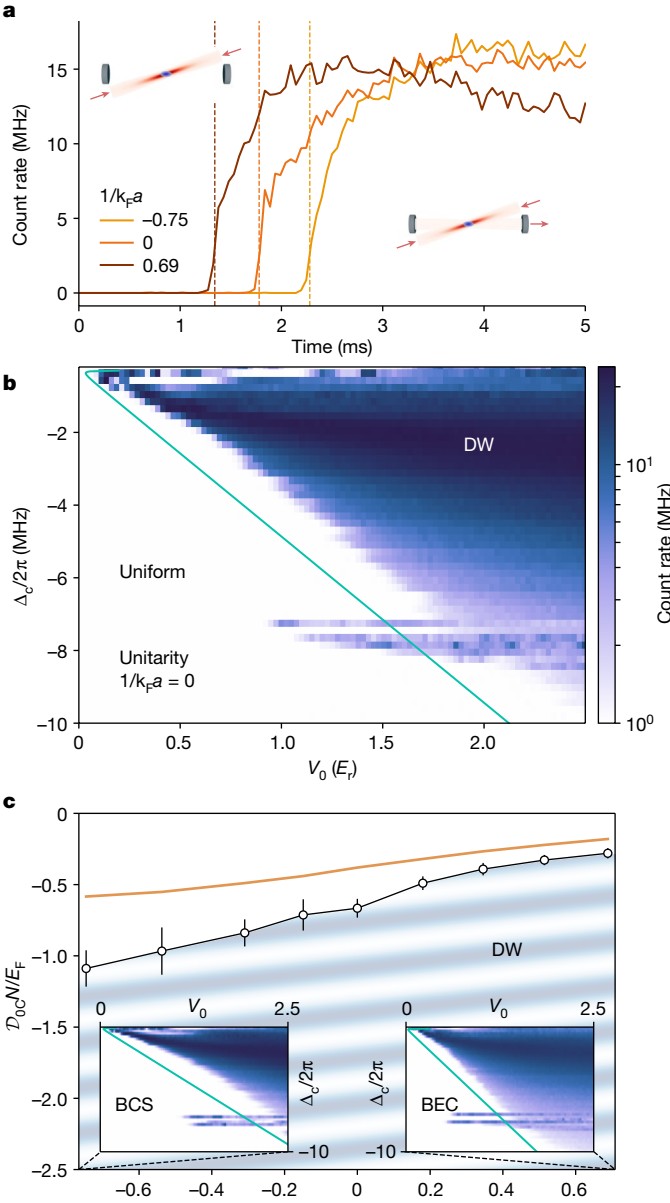

**Fig. 1 | Concept of the experiment. a**, A strongly interacting Fermi gas trapped inside a high-finesse optical resonator is illuminated by a standing-wave pump laser with wave vector $\mathbf{k}_p$, polarized along the direction of the magnetic field B, which intersects the axis of the cavity mode ($x$ direction) with wave vector $\mathbf{k}_c$ at an angle of 18°. The pump beam couples dispersively to atomic motion. Off-resonant scattering of pump photons by the atoms into the cavity mode and vice versa leads to an effective infinite-range interaction between atoms. Above a critical strength, the infinite-range interaction results in a superradiant phase transition to a DW-ordered state with spatial modulation at $2\pi/\mathbf{k}_-$. **b**, In the left panel, photon scattering from the pump into the cavity and vice versa via the atoms imparts momentum kicks $\mathbf{k}_\pm = \mathbf{k}_c \pm \mathbf{k}_p$ onto the latter, displacing the Fermi surface. In the right panel, since $|\mathbf{k}_-| < \mathbf{k}_F$, the photon-mediated interactions induce particle-hole excitations at the Fermi surface in addition to Cooper pairing arising from the contact interactions.

dimensions to cavity-induced superconducting pairing and topological states have been predicted for fermions[24–32].

Here, we realize a doubly tunable Fermi gas combining simultaneously and independently the control over contact and photon-mediated long-range interactions. We explore the regime where both interactions are strong, the latter leading to DW ordering. For fermionic particles, the Pauli principle restricts the effects of interactions to the Fermi surface; thus, the resonant s-wave contact interactions yield Cooper pairing at low temperatures. By contrast, the photon-mediated interaction couples particle-hole excitations on the Fermi surface at discrete wave vectors $\mathbf{k}_\pm = \mathbf{k}_c \pm \mathbf{k}_p$, imposed by the pump-cavity geometry as illustrated in Fig. 1a. In our three-dimensional system, the low-energy physics is associated with scattering processes with the wave vector $\mathbf{k}_-$, which is smaller than the Fermi wave vector $\mathbf{k}_F$, leading to a broad particle-hole spectrum (in contrast to ref. 18). This is described by the Lindhard function for free fermions, which is maximum at zero frequency for low momenta close to $\mathbf{k}_-$. This contrasts with large momenta, where the Pauli principle does not restrict the available phase space unless the Fermi surface is deformed[18]. We find that even in the presence of strong contact interactions, photon-mediated interactions modify the zero-frequency particle-hole susceptibility and lead to the spontaneous formation of a DW pattern above a critical strength in the attractive case.

In the experiment, we prepare a degenerate Fermi gas of $N = 3.5 \times 10^5$ Li atoms equally populating the two lowest hyperfine states, trapped within a mode of a high-finesse optical cavity[33,34] and in the vicinity of a broad Feshbach resonance at 832 G. We turn on the photon-mediated interactions by illuminating the cloud from the side using a retro-reflected pump beam. The pump and the neighbouring cavity resonance are detuned with respect to the atomic $D_2$ transition by $-2\pi \times 138.0$ GHz. There, the atoms induce a dispersive shift of the cavity

**Fig. 2 | Phase diagrams of the system. a**, Photon traces recorded at fixed $\Delta_c = -2\pi \times 2$ MHz as a function of the linearly increasing pump strength $V_0$ for different values of the short-range interaction parameter $1/k_F a$ spanning the strongly interacting regime of the BCS–BEC crossover. Each measurement features a sharp increase of the photon count rate above a critical value of the pump strength $V_{0C}$ (dashed vertical lines). **b**, Phase diagram of the unitary Fermi gas in the $V_0$–$\Delta_c$ plane, exhibiting DW self-ordering. The solid line is a theory estimate of the phase boundary (in the text). **c**, Measurement of the critical long-range interaction strength $\mathcal{D}_{0C}$ as a function of the contact interaction parameter at fixed $\Delta_c = 6\delta_c$. Above the critical value, the system exhibits a modulated density, depicted by the oblique stripes. The solid line is the critical interaction strength calculated from theory. Insets display phase diagrams measured in the BCS and BEC regimes for the same parameter range as the one of **b**. Error bars represent standard deviations.

resonance by $\delta_c = U_0 N/2 = -2\pi \times 280$ kHz, exceeding the cavity line width $\kappa_c = 2\pi \times 77(1)$ kHz. The pump beam intersects the cavity at an angle of 18°, such that two discrete density-fluctuation modes at momenta $\mathbf{k}_\pm$ are coupled to light, as illustrated in Fig. 1b. The low incidence angle results in the hierarchy $|\mathbf{k}_-| \ll |\mathbf{k}_+|$, so that only the mode at $\mathbf{k}_-$ contributes to the low-energy physics (Methods). We use pump-cavity detunings

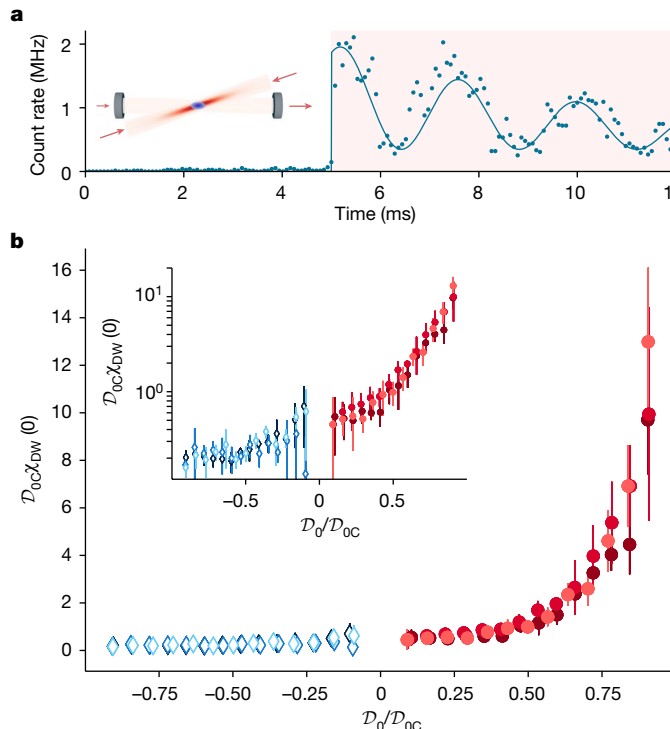

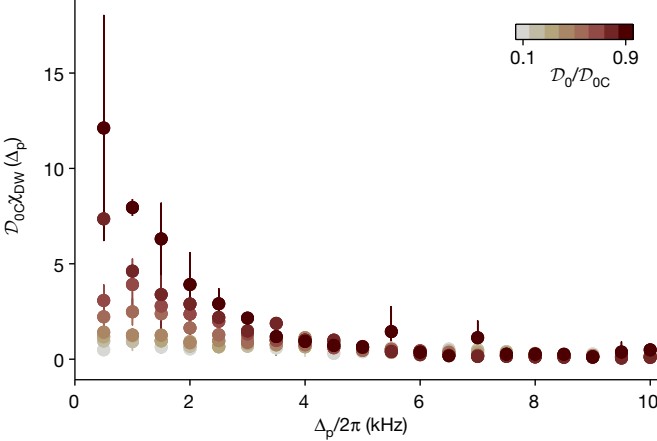

**Fig. 4 | Measurement of the DW response $\chi_{DW}(\Delta_p)$ of a unitary Fermi gas as a function of $\Delta_p$ and for values of $\mathcal{D}_0/\mathcal{D}_{0C}$ between 0.1 and 0.9.** The absence of structure at finite frequency confirms the absence of mode softening. The data are taken for $\Delta_c = -2\pi \times 2$ MHz. Error bars represent standard deviations.

**Fig. 3 | Zero-frequency DW susceptibility $\chi_{DW}(0)$ measurement. a**, Photon trace acquired while a weak on-axis probe beam is sent inside the cavity after the pump strength has been ramped over 5 ms to a value below the critical one. The solid line is a fit to the data (Methods), from which we extract the zero-frequency DW susceptibility $\chi_{DW}(0)$. The shaded area highlights the interval during which the probe is on. **b**, Measured DW susceptibility as a function of the long-range interaction strength below the critical value for both attractive (red dots) and repulsive (blue diamonds) long-range interactions and for three different values of the contact interaction parameter ($1/\mathbf{k}_F a = -0.75$, 0 and 0.69 from light to dark). The measurements were performed at constant absolute detuning $|\Delta_c - \delta_c| = 2\pi \times 1.7$ MHz. In the inset, the same data are displayed in logarithmic scale. Error bars represent standard deviations.

$|\Delta_c|/2\pi$ between 1 and 10 MHz for which $|\Delta_c| \gg |\delta_c|, \kappa_c$, and the cavity field adiabatically follows the atomic dynamics, ensuring that the system is accurately described by the Hamiltonian (equation (1)).

## DW ordering

We observe DW ordering upon increasing the strength of the photon-mediated interaction above a critical threshold. Experimentally, at fixed scattering length, we linearly ramp up the pump power and monitor the intracavity photon number by recording the photon flux leaking through one of the cavity mirrors while keeping all other parameters fixed. In Fig. 2a, we show typical photon traces for different scattering lengths, as $V_0$ is linearly increased up to $2.5\,E_r$ over 5 ms, with $E_r = \hbar^2\mathbf{k}_c^2/2m = h \times 73.67$ kHz the recoil energy. The build-up in the cavity field above a critical pump strength $V_{0C}$ marks the onset of DW ordering (Methods).

Repeating this measurement as a function of $\Delta_c$, we construct the phase diagram of the system in the $V_0$–$\Delta_c$ plane, presented in Fig. 2b for the unitary gas. For small $|\Delta_c|$, the phase boundary is a straight line, corresponding to a constant ratio $V_0/\Delta_c$, showing that the boundary is determined only by $\mathcal{D}_0$. For $|\Delta_c| \lesssim |\delta_c|$, we observe instabilities likely due to optomechanical effects. For $|\Delta_c| > 2\pi \times 3$ MHz, we observe a systematic deviation from the linearity, probably due to the lattice formed by the pump, changing the gas properties[35]. This single-particle effect is not captured by the effective interaction Hamiltonian (equation (1)).

The structures arising at $\Delta_c \approx -2\pi \times 7$ MHz and $-2\pi \times 8$ MHz originate from the presence of high-order transverse modes of the cavity, with mode functions overlapping with the atomic density[33].

We acquire similar phase diagrams at different scattering lengths and find a transition to the DW-ordered phase for sufficiently strong pumps throughout the entire BEC and Bardeen–Cooper–Schrieffer superfluid (BCS) crossover. While the phase diagrams are qualitatively similar, with a linear phase boundary at small $\Delta_c$, we observe a systematic shift of the DW phase boundary toward larger pump strengths as the system crosses over from the BEC to the BCS regime. In the regime $0.7$ MHz $< \Delta_c/2\pi < 3$ MHz, the linear phase boundary observed at unitarity persists for all scattering lengths. This allows us to describe the DW self-ordering transition in terms of the single long-range interaction parameter $N\mathcal{D}_0/E_F$. Figure 2c presents the phase diagram in the parameter plane of the short-range versus long-range interaction strength. We observe a smooth dependence of the phase boundary on the short-range interaction, with a systematically lower critical long-range interaction strength in the BEC side.

To understand this phase diagram, we start from the critical point $\mathcal{D}_{0C} = -1/2\chi_0$, expected from the mean-field and random-phase approximations applied to the long-range interaction (Methods). Here, $\chi_0$ is the zero-frequency susceptibility of the gas in the absence of the long-range interaction. To predict quantitatively the phase boundary in the BCS–BEC crossover, we disregard the effects of the pump lattice and the contribution of the density response at $\pm\mathbf{k}_+$ and approximate $\chi_0$ by its long-wavelength limit, the compressibility. The latter is obtained from accurate measurements of the equation of state as a function of the scattering length[36,37]. The resulting predictions for the phase boundary are presented as solid lines in Fig. 2b,c. This simple, parameter-free theory captures very well the relative changes of the critical point across the crossover (Extended Data Fig. 3). It, however, underestimates the absolute threshold by approximately a factor of two for all short-range interaction strengths, indicating that the zero-temperature compressibility overestimates the actual susceptibility. We indeed expect that finite wave vector and finite temperature should generally decrease the susceptibility.

## Susceptibility measurement

While the measurement of the cavity field allows for the identification of the onset of DW order, it does not yield information on the photon-mediated interactions below the transition. Nevertheless, the long-range interactions strongly modify properties of the gas even

far below the ordering transition via virtual cavity photons. We now explore this by directly measuring the DW response function $\chi_{DW}(\omega)$ as a function of the long- and short-range interaction strengths. To this end, we drive the cavity on axis using a very weak probe laser in addition to the transverse pump[38], imposing a DW pattern at $\mathbf{k}_\pm$. The resulting photon-leakage rate yields $\chi_{DW}$ from the linear response theory (Methods).

In practice, the atomic response depends on the relative phase of the pump and the probe. This is intimately connected to the underlying $\mathbb{Z}_2$ symmetry of the model, which is broken in the ordered phase, as observed in earlier experiments on BECs[13,39]. We circumvent this issue by introducing a small detuning $\Delta_p$ between the pump and the probe, such that the phase winds adiabatically during the probing time, leading to slowly oscillating intracavity photon numbers. In the limit $\Delta_p \to 0$, the amplitude of the oscillations observed in an experimental realization provides a direct measure of the zero-frequency DW response function $\chi_{DW}(0)$ (Methods).

Experimentally, we first fix the long- and short-range interaction strengths by, respectively, fixing the pump power and offset magnetic field, and then, we shine the probe for 10 ms with $\Delta_p = 2\pi \times 200$ Hz. A typical signal is shown in Fig. 3a for $\Delta_c = -2\pi \times 2$ MHz and $V_0 = 0.75\ E_r$, exhibiting the expected oscillations at $2\Delta_p$ together with damping, likely due to heating resulting from the large oscillating signal. The amplitude of the initial oscillation can be directly fitted to yield the value of $\chi_{DW}(0)$. For attractive photon-mediated interactions, the intracavity photon number is strongly enhanced by the presence of the atoms, as the gas coherently transfers photons from the pump to the cavity, similar to an optical parametric amplifier.

In Fig. 3b, we show the measured values of $\chi_{DW}(0)$ for $\mathcal{D}_0$ up to 0.9 $\mathcal{D}_{0C}$ at $\Delta_c = 5\delta_c < 0$ and $1/\mathbf{k}_F a = -0.75$, 0 and 0.69 (red dots). We observe an increase of the susceptibility over more than one order of magnitude with increasing $\mathcal{D}_0$, which is the expected feature of second-order phase transitions. This was observed for self-organization and supersolid transitions in non-interacting BECs[38,40]. For repulsive photon-mediated interactions ($\Delta_c > 0$, blue diamonds), no ordering is expected or observed, and we observe a reduction of the susceptibility by up to a factor of approximately three over the same range of $|\mathcal{D}_0|$. Up to normalization of $\chi_{DW}(0)$ and $\mathcal{D}_0$ by $\mathcal{D}_{0C}$, we observe that for attractive or repulsive long-range interactions, the variations of the susceptibility are identical within error bars for all scattering lengths in the BCS–BEC crossover. This highlights the versatility of our system in independently tuning the short- and long-range interactions, therefore addressing separately pairing and particle-hole channels.

The attractive (repulsive) photon-mediated interactions lower (raise) the energy cost of particle-hole excitations. For bosons with a sharp single-frequency excitation spectrum, this leads to a mode softening of the corresponding excitation mode, touching zero at the critical point[38,41]. Free fermions at low momenta, in contrast, feature a continuous, incoherent gapless particle-hole spectrum[42], such that no soft mode is expected.

We now investigate this effect for a strongly interacting Fermi gas by extending our susceptibility measurements to finite frequencies by systematically scanning $\Delta_p$ up to $2\pi \times 10$ kHz, larger than $\hbar^2 \mathbf{k}_\perp^2/2m = h \times 7.2$ kHz, the recoil energy associated with $\mathbf{k}_\perp$. We then extract $\chi_{DW}(\Delta_p)$ from the amplitude of the photon trace oscillations at $2\Delta_p$. For the unitary Fermi gas, the results are presented in Fig. 4 for $\mathcal{D}_0$ up to $0.9\mathcal{D}_{0C}$, all showing that $\chi_{DW}(\Delta_p)$ monotonically decreases with frequency $\Delta_p$. The low-frequency susceptibility increases upon approaching the transition, while the higher-frequencies parts of the spectrum remain unchanged. We observe such a behaviour for all accessible scattering lengths in the BCS–BEC crossover. This contrasts with the mode softening observed with weakly interacting BECs. While this would be expected in our geometry for free fermions, due to the broad particle-hole spectrum, it is surprising that this feature is also present for the unitary Fermi gas, which is known to also display a phonon spectrum at low momentum[43,44]. This might be due to the strongly interacting nature of the system leading to the damping of the excitations but could also originate from the combination of finite temperature and trap averaging.

## Discussion

We operate with atoms in the deeply degenerate regime with temperatures on the order of $T = 0.08\ T_{Fh}$, with $T_{Fh}$ the Fermi temperature calculated for a harmonic trap, where for all interaction strengths, the system is superfluid in the absence of the photon-mediated interactions. For a wide range of the short-range interaction strength, the system enters the DW-ordered phase upon increasing the photon-mediated interaction strength and returns to the superfluid phase when the long-range interaction is ramped back to zero, with limited heating (Extended Data Fig. 1). However, this leaves open the fascinating question of whether the system remains paired and superfluid in the presence of strong long-range interactions and in the DW-ordered state.

Compared with condensed-matter systems showing an interplay of charge DW and superfluidity[45], our system has a fully controllable microscopic Hamiltonian. The photon-induced DW order shares similarities with type II charge-DW compounds[46], with cavity photons playing the role of phonons in real materials. In this context, the real-time weakly destructive measurement channel through the cavity field opens the possibility of gaining insight into the interplay of structural effects and strong interactions in complex quantum materials.

Our platform complements ongoing research in the field of cavity-coupled strongly correlated materials, where the cavity photons couple to the kinetic energy of charges through the Peierls phase[47,48] or indirectly via interband transitions or collective modes. Interestingly, a direct two-photon density coupling similar to ours has been predicted for side-pumped two-dimensional materials, originating from diamagnetic interactions between charges and light and leading to enhanced superconductivity[49].

Natural extensions of our experiment include the use of several pumping frequencies addressing multiple cavity modes, providing further control over the long-range interaction potential[12,23], and the study of retardation effects due to our cavity line width being comparable with the photon recoil energy at $k_c$ (ref. 15). A fascinating perspective is to operate the pump in the vicinity of a photo-association transition[50], offering the possibility to induce long-range pair–pair interactions.

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

## Methods

### Experimental procedure

We produce a strongly interacting Fermi gas of $^6$Li following the method described in refs. 33,34. This procedure produces deeply degenerate, balanced mixtures of the two lowest hyperfine states trapped in a crossed dipole trap elongated along the cavity axis, formed by two Gaussian laser beams with waists of 33 μm intersecting each other with an angle of 36°.

Thermometry is performed by releasing the cloud into a hybrid trap, formed by one of the arms of the dipole trap and the residual curvature of the magnetic field[34]. An in situ absorption image is then taken with a light intensity optimized for the signal-to-noise ratio, and the density profile is obtained from the image using finite-saturation corrections. The reduced temperature in this trap is deduced from the shape of the cloud at unitarity. This yields a $T/T_{Fh}$ with $T_{Fh} = \hbar\bar{\omega}(3N)^{1/3}$, with $N$ the total number of atoms and $\bar{\omega} = (\omega_x\omega_y\omega_z)^{1/3} = 2\pi \times 106$ Hz is the geometric mean of the oscillation frequencies in the hybrid trap. This provides us with an upper bound of the degree of degeneracy in the crossed dipole trap.

The hybrid trap is harmonic and allows for both precise thermometry and calibration of each beam geometry. To reach the lowest temperatures, we found out that the crossed dipole trap operates in a regime where the anharmonicity is too strong to allow for harmonic approximation. For the purpose of evaluating the theoretical phase boundary, we instead use the full crossed-Gaussian beam trap shape deduced from trap frequencies measured in each beam separately. We then deduce the density distribution using the zero-temperature equation of state in the BEC–BCS crossover[36,37].

The pump beam is linearly polarized along the magnetic-field direction, and we estimate its waist to be 120 μm, much larger than the Thomas–Fermi radii of the cloud. We calibrate the depth of the pump lattice using Kapitza–Dirac diffraction on a molecular BEC at $B = 695$ G (ref. 51). The photons leaking from one of the cavity mirrors are detected using a single-photon counting module with an efficiency of approximately 3% (ref. 52).

### Heating due to the side pumping

We estimate the heating due to the pump by measuring the temperature of the cloud after linearly ramping up the pump lattice depth to varying end values at a constant rate and then, ramping it back to zero with the same rate. With increasing pump power, we observe a monotonically increasing temperature of the cloud shown in Extended Data Fig. 1. Interestingly, temperature shows no particular feature when the pump power reaches and exceeds the DW-ordering threshold. At the critical point, we measure a temperature of $T = 0.12(2)T_{Fh}$, an increase by a factor of 50% compared with the initial one. Heating is sufficient to heat the cloud above the superfluid critical temperature of $0.21T_{Fh}$ (ref. 53) for a strength of the long-range interactions exceeding $2\mathcal{D}_{0C}$, deep in the ordered phase. By extracting the atom number from the density profiles, we verify that the losses display the same trend with varying pump strength.

### Theoretical model

The Fermi gas is coupled to a single standing-wave mode designated by the operator $\hat{a}$ of the cavity with the single atom-photon coupling strength $g(\mathbf{r}) = g_0\cos(\mathbf{k}_c\cdot\mathbf{r})$, where $\mathbf{k}_c = |\mathbf{k}_c|\mathbf{e}_x = k_c\mathbf{e}_x$ is the cavity wave vector. The atomic cloud is also transversely pumped by an incident, back-reflected pump laser with the wave vector $\mathbf{k}_p$, where $k_p = |\boldsymbol{k}_p| \simeq k_c$ and frequency $\omega_p = ck_p$. In the dispersive regime, the atoms experience an effective lattice potential[11], identical for the two hyperfine components of the gas:

$$\hat{V}_{latt}(\mathbf{r}) = V_0\cos^2(\mathbf{k}_p\cdot\mathbf{r}) + U_0\hat{a}^\dagger\hat{a}\cos^2(\mathbf{k}_c\cdot\mathbf{r}) + \eta_0(\hat{a} + \hat{a}^\dagger)\cos(\mathbf{k}_p\cdot\mathbf{r})\cos(\mathbf{k}_c\cdot\mathbf{r}),$$

(2)

where $\eta_0 = \sqrt{V_0U_0}$. This potential is added to the external trap potential $V_{tr}(\mathbf{r})$.

In the frame rotating at the pump-laser frequency, the system is described by the Hamiltonian (we set $\hbar = 1$ throughout this section),

$$\hat{H} = -\Delta_c\hat{a}^\dagger\hat{a} + \beta[\hat{a}e^{-i(\Delta_p t - \phi_0)} + \text{H.c.}]$$
$$+ \sum_{\sigma=\downarrow,\uparrow} \times \int d\mathbf{r}\hat{\Psi}_\sigma^\dagger(\mathbf{r})\left[-\frac{\nabla^2}{2m} - \mu_\sigma + V_{tr}(\mathbf{r}) + \hat{V}_{latt}(\mathbf{r})\right]\hat{\Psi}_\sigma(\mathbf{r})$$
$$+ \int d\mathbf{r}d\mathbf{r}'\hat{\Psi}_\uparrow^\dagger(\mathbf{r})\hat{\Psi}_\downarrow^\dagger(\mathbf{r}')V_{sr}(\mathbf{r} - \mathbf{r}')\hat{\Psi}_\downarrow(\mathbf{r}')\hat{\Psi}_\uparrow(\mathbf{r}),$$

(3)

where the first term is the free cavity Hamiltonian with the pump-cavity detuning $\Delta_c = \omega_p - \omega_c$, $\hat{\Psi}_\sigma(\mathbf{r})$ is the fermionic annihilation-field operator for spin $\sigma = \{\downarrow, \uparrow\}$, $\mu_\sigma$ is the chemical potential, and $V_{sr}(\mathbf{r} - \mathbf{r}')$ is a pseudopotential yielding the s-wave scattering length $a$ between two atoms[54]. For later use, we have also included an on-axis probe with strength $\beta$, the pump-probe detuning $\Delta_p = \omega_p - \omega_{probe}$ and an initial phase $\phi_0$. In the experiment, $\beta = 0$, except for the purpose of measuring the DW response function $\chi_{DW}(\omega)$ (see the main text and Linear response theory and the DW response function χDW(ω)).

The Hamiltonian equation (3) can be recast in the form

$$\hat{H} = \hat{H}_{at} - \tilde{\tilde{\Delta}}_c\hat{a}^\dagger\hat{a} + \eta_0(\hat{a} + \hat{a}^\dagger)\hat{\Theta}$$
$$+ \beta[\hat{a}e^{-i(\Delta_p t - \phi_0)} + \text{H.c.}],$$

(4)

where $\hat{H}_{at}$ is the Hamiltonian of an interacting, trapped two-component Fermi gas with a classical lattice potential $V_p(\mathbf{r}) = V_0\cos^2(\mathbf{k}_p\cdot\mathbf{r})$ formed by the pump. Here, $\tilde{\tilde{\Delta}}_c = \Delta_c - \hat{\delta}_c = \Delta_c - U_0\int d\mathbf{r}\cos^2(\mathbf{k}_c\cdot\mathbf{r})\hat{n}(\mathbf{r})$, with $\hat{n}(\mathbf{r}) = \sum_\sigma \hat{n}_\sigma(\mathbf{r}) = \sum_\sigma \hat{\Psi}_\sigma^\dagger(\mathbf{r})\hat{\Psi}_\sigma(\mathbf{r})$ being the total density operator, is the dispersively shifted pump-cavity detuning and

$$\hat{\Theta} = \int d\mathbf{r}\cos(\mathbf{k}_p\cdot\mathbf{r})\cos(\mathbf{k}_c\cdot\mathbf{r})\hat{n}(\mathbf{r})$$
$$= \frac{1}{4}(\hat{n}_{\mathbf{k}_+} + \hat{n}_{-\mathbf{k}_+} + \hat{n}_{\mathbf{k}_-} + \hat{n}_{-\mathbf{k}_-}),$$

(5)

with $\hat{n}_\mathbf{q} = \int d\mathbf{r}\hat{n}(\mathbf{r})e^{i\mathbf{q}\cdot\mathbf{r}}$ being the Fourier component of the total density operator, is the atomic DW operator describing the modulation of the atomic density at wave vectors $\mathbf{k}_\pm = \mathbf{k}_p \pm \mathbf{k}_c$.

In the Hamiltonian equation (4) describing our experiment, $\Delta_c$ is much larger than all other energy scales (including the dispersive shift $\delta_c = \langle\hat{\delta}_c\rangle$, so that $\tilde{\Delta}_c = \langle\tilde{\tilde{\Delta}}_c\rangle \simeq \Delta_c$, so that the cavity-field dynamics is very fast and follows the atomic dynamics. The steady-state cavity-field operator can, therefore, be obtained through the Heisenberg equation of motion, yielding

$$\hat{a} = \frac{1}{\Delta_c + i\kappa_c}\left[\eta_0\hat{\Theta} + \beta e^{i(\Delta_p t - \phi_0)}\right].$$

(6)

Substituting the steady-state cavity-field operator (6) in the Hamiltonian (4) and ignoring a constant term yields an effective, atom-only description of the system (up to the inverse square of the detuning of the pump laser with respect to the atomic transition)[11]:

$$\hat{H}_{eff-at} = \hat{H}_{at} + \mathcal{D}_0\hat{\Theta}^2 + \frac{2\beta}{\eta_0}\mathcal{D}_0\hat{\Theta}\cos(\Delta_p t - \phi_0),$$

(7)

where $\mathcal{D}_0 = \Delta_c\eta_0^2/(\Delta_c^2 + \kappa_c^2) \simeq \eta_0^2/\Delta_c$ is the strength of the cavity-mediated long-range density–density interaction. In the last equality, we asserted $\kappa_c \ll \Delta_c$, as realized in the experiment. The last term in equation (7) is

the driving of the Fermi gas due to the interference between the pump and the on-axis probe.

## Theoretical phase boundary

We identify the critical pump threshold $\eta_{0C} = \sqrt{V_{0C}U_0}$ that separates the superradiant phase from the normal state through perturbation theory[55] by integrating out the atomic degrees of freedom and expanding the resultant free energy in powers of the order parameter $\hat{\Theta}$. Up to second order in the order parameter, we obtain the free energy as in Landau theory,

$$F \approx (\eta_{0C}^2 - \eta_0^2)\hat{\Theta}^2 + O(\hat{\Theta}^4), \tag{8}$$

where $\eta_{0C}^2 = -(\Delta_c^2 + \kappa_c^2)/2\Delta_c\chi_0 \simeq -\Delta_c/2\chi_0$. This corresponds to the critical long-range interaction strength $\mathcal{D}_{0C} = -1/2\chi_0$, where $\chi_0$ denotes the atomic susceptibility representing the response of the interacting Fermi gas to density perturbations at the wave vectors $\mathbf{k}_\pm$ in the absence of the pump and cavity lattices:

$$\chi_0 = \frac{1}{16}\sum_{\mathbf{q}=\pm\mathbf{k}_\pm}\chi_0^R(\mathbf{q}). \tag{9}$$

Here, $\chi_0^R(\mathbf{q})$ is the retarded density–density response function at zero frequency and wave vector $\mathbf{q}$, calculated at a fixed finite scattering length. It coincides with the Lindhard function for a non-interacting Fermi gas.

To compare with the experiment, we first note that the short-wavelength contributions to $\chi_0$ at $\pm\mathbf{k}_+$ are negligible compared with the low momentum one. Indeed, for $|\mathbf{k}_+| \gg k_F$, the density response can be evaluated in the BCS–BEC crossover using operator product expansion[52], yielding to lowest order $\chi_0^R(\mathbf{k}_+) \approx 2N/\epsilon_{\mathbf{k}_+}$ with $\epsilon_{\mathbf{k}_+} = \hbar^2\mathbf{k}_+^2/2m$. Throughout the BCS–BEC crossover, the ratio $\chi_0^R(\mathbf{k}_-)/\chi_0^R(\mathbf{k}_+)$ is the smallest in the far-BCS regime and bounded from below by $3\epsilon_{\mathbf{k}_+}/4E_F$, which is approximately 12 for our parameters.

We then evaluate the long-wavelength contributions $\chi_0^R(\mathbf{q} = \pm\mathbf{k}_-)$. For $\mathbf{q} \to 0$, the compressibility sum rule gives $\chi_0^R(0) = \partial n/\partial\mu = n^2\kappa$, with $\kappa$ being the compressibility. For low but finite $\mathbf{q} = \pm\mathbf{k}_-$, hydrodynamics is expected to provide a good description of the density response, which suggests that $\chi_0^R(\mathbf{q})$ is essentially independent of momentum[56]. We therefore use the compressibility $\kappa$ inferred from the thermodynamic equation of state as an estimate of $\chi_0^R(\pm\mathbf{k}_-)$ in the BCS–BEC crossover. The equation of state of a homogeneous Fermi gas has been measured accurately as a function of the contact interaction strength[36,37]. We use the interpolation formula for the universal thermodynamic functions provided in ref. 36 to deduce the compressibility of the homogeneous Fermi gas. We then use the local density approximation to perform trap averaging and to relate it to the Fermi energy $E_F$ at the centre of the trap.

## Linear response theory and the DW response function $\chi_{DW}(\omega)$

We now turn our attention to the last term of equation (7) arising from the on-axis pumping of the cavity mode. We calculate the response of the DW order operator to first order using the Kubo formula,

$$\begin{aligned}\langle\hat{\Theta}(t)\rangle = \langle\hat{\Theta}\rangle_0 \\ + \frac{2\beta\mathcal{D}_0}{\eta_0}\int_{-\infty}^{\infty}dt'\chi_{DW}(t-t')\cos(\Delta_p t'-\phi_0),\end{aligned} \tag{10}$$

where the DW response function $\chi_{DW}(t-t')$ is given by

$$\chi_{DW}(t-t') = -i\theta(t-t')\langle[\hat{\Theta}(t),\hat{\Theta}(t')]\rangle_0. \tag{11}$$

Here, $\theta(t)$ is the unit step function and $\langle\ldots\rangle_0$ implies averaging with $\beta = 0$.

Introducing the Fourier transform $\chi_{DW}(\Delta_p) = \int_{-\infty}^{\infty}d\tau\chi_{DW}(\tau)e^{-i\Delta_p\tau}$ and noting that $\chi_{DW}(\Delta_p) = \chi_{DW}^*(-\Delta_p)$, equation (10) can be recast as

$$\delta\langle\hat{\Theta}(t)\rangle = \frac{2\beta\mathcal{D}_0}{\eta_0}\text{Re}\left[\chi_{DW}(\Delta_p)e^{i(\Delta_p t-\phi_0)}\right], \tag{12}$$

where $\delta\langle\hat{\Theta}(t)\rangle \equiv \langle\hat{\Theta}(t)\rangle - \langle\hat{\Theta}\rangle_0$. In the low-frequency limit $\Delta_p \ll c_s|\mathbf{k}_-|$, where $c_s$ is the speed of sound, the dynamical response function is purely real and $\chi_{DW}(\Delta_p) \simeq \chi_{DW}(0) + O((\Delta_p/c_s|\mathbf{k}_-|)^2)$ such that we obtain

$$\delta\langle\hat{\Theta}(t)\rangle \simeq \frac{2\beta\mathcal{D}_0}{\eta_0}\chi_{DW}(0)\cos(\Delta_p t-\phi_0). \tag{13}$$

Below the superradiant threshold, $\langle\hat{\Theta}\rangle_0 = 0$, and the intracavity photon signal to first order then reads

$$\langle\hat{a}^\dagger\hat{a}\rangle = \frac{\beta^2}{\Delta_c^2}[1+4\mathcal{D}_0\chi_{DW}(0)\cos^2(\Delta_p t-\phi_0)], \tag{14}$$

relating the oscillation in the intracavity photon number to the DW susceptibility $\chi_{DW}(0)$.

## Data analysis

The value of the critical pump depth $V_{0C}$ at which the system undergoes the phase transition is inferred from photons leaking out of the cavity while the pump depth is increased. For a single realization of the experiment, we construct the histogram of arrival times of photons on the detector as a function of the pump depth, which increases linearly with time. Then, $V_{0C}$ is determined from the point at which the slope of the reconstructed photon trace is the highest, obtained from taking its numerical derivative.

We extract $\chi_{DW}(0)$ from a fit of measured photon traces to the model described by equation (14). We account for the amplitude decay of the oscillation through the addition of a factor $e^{-t/\tau}$ to the oscillatory term of the model. This may in particular capture heating and atomic losses during the measurement. Interestingly, the damping factor $1/\tau$ of the measured response features a continuous increase as the pump power approaches the threshold, as shown in Extended Data Fig. 2. The phase offset $\phi_0$ is distributed uniformly over $[0,\pi]$ for different realizations, as expected for a random relative phase between the pump and the probe. We verified that for all values of pump power, the fitted amplitude of the response varies linearly with the probe power, validating the linear response hypothesis underlying the fit.

## Data availability

All data files are available from the corresponding author upon request. Accompanying data, including those for figures, are available from the Zenodo repository (https://zenodo.org/record/7733304).

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

**Acknowledgements** We acknowledge discussions with T. Donner and T. Esslinger. We thank G. del Pace and T. Bühler for their assistance in the final stages of the experiment. We acknowledge funding from the European Research Council under the European Union Horizon 2020 Research and Innovation Programme (Grant no. 714309) and the Swiss National Science Foundation (Grant no. 184654). F.M. acknowledges financial support from the Austrian Science Fund (Stand-Alone Project P 35891-N).

**Author contributions** V.H., T.Z., K.R. and H.K. performed experiments. V.H. and T.Z. processed the data. F.M., E.C. and H.R. performed calculations. J.-P.B. planned and supervised the experiments.

**Funding** Open access funding provided by EPFL Lausanne.

**Competing interets** The authors declare no competing interests.

**Additional information**
**Correspondence and requests for materials** should be addressed to Jean-Philippe Brantut.

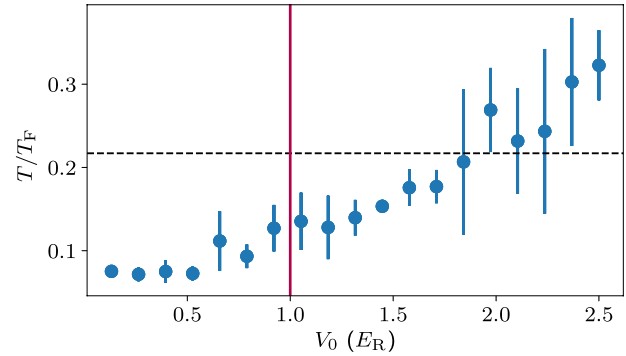

**Extended Data Fig. 1 | Measurement of heating due to the pump beam.**
The vertical line depicts the location of threshold for the self-organizing phase transition and the horizontal dashed one marks the superfluid transition for a homogeneously trapped unitary fermi gas. As the pump power is increased, we observe a smooth increase of the gas temperature showing no dramatic behavior around the self-organization phase transition. Error bars represent a standard deviation.

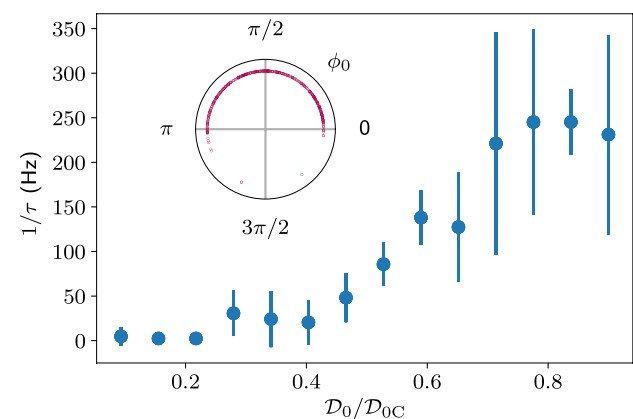

**Extended Data Fig. 2 | Measured damping of the oscillatory behavior predicted by Eq. (14), which is accounted for by an additional $e^{-t/\tau}$ factor in the equation.** The signal becomes strongly damped as the critical value for the long-range interaction strength is approached. The data shown is part of the set displayed in Fig. 3 of the main text, here taken at unitarity and for $\Delta_c < 0$. In inset, we display the measured phase offset $\phi_0$ which features a uniform distribution. Error bars represent a standard deviation.

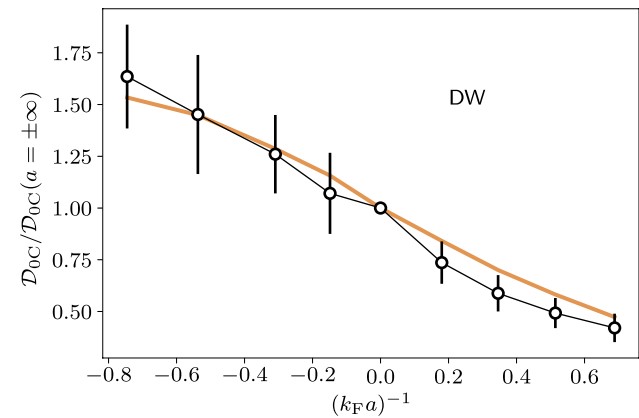

**Extended Data Fig. 3 | Critical long-range interaction strength as a function of short-range interaction strength, normalized with respect to the critical strength at unitarity (black circles), compared with the theoretical prediction based on the compressibility (solid orange line).** The data are identical to that of Fig. 2c. Error bars represent a standard deviation.