## [Peer Review File · Nature]

Manuscript Title: Density-wave ordering in a unitary Fermi gas with photon-mediated interactions

Reviewer Comments & Author Rebuttals

Reviewer Reports on the Initial Version:

Referees' comments:

Referee #1 (Remarks to the Author):

In the manuscript "Density-wave ordering in a unitary Fermi gas with photon-mediated interactions," the authors present a tunable experimental setup where they combine both contact interaction and long-ranged photon-mediated interactions in a Fermi gas. Thanks to the independent tunability of the two types of interactions, they find the onset of density-wave ordering, which they identify by measuring the Dicke-like normal-to-superradiant photon emission. Moreover, because they can address both types of interactions separately, they can explore the entire phase diagram as a function of the contact interaction and the long-range mediated interaction (both attractive and repulsive), which they benchmark via a mean-field theory. Finally, they study the properties of the photon-mediated interactions by measuring the zero-temperature on-shell susceptibility.

In my opinion, the manuscript is well-written, presents the results and methodology clearly, includes appropriate bibliography and robust data for their observations, and introduces exciting avenues for studying tunable strong light-matter interactions in strongly-correlated systems. In this way, they make substantial progress in the field of quantum gases and open new questions about tailoring exotic quantum many-body states such as long-range pairing. Other phenomena appear in their data, such as the structures in the phase diagram, but the authors provide reasonable arguments to explain them. My only concern is about the limitations of using a mean-field theory to estimate the phase boundary via the density-density response function. The authors mention that the discrepancies may be due to finite temperature effects. However, while a mean-field theory offers a good benchmark, a many-body approach seems necessary to include other important effects (e.g., retardation to the response function or the same finite temperature behavior). It would be nice if the authors added a comment in this direction.

In any case, I believe the manuscript meets the requirements of novelty and originality demanded by Nature and deserves to be published as a regular article.

Referee #2 (Remarks to the Author):

The manuscript, "Density-wave ordering in a unitary Fermi gas with photon-mediated interactions" by Helson et al., presents a study of interacting degenerate Fermi gases coupled to cavities and the density wave instability that ensues. Most notably, they present interesting data as a function of

scattering sign and strength and, in fig 4, present interesting data showing the DW response versus probe frequency. This is a new contribution. Overall, the manuscript is very well written and presents a solid result that will open new avenues of inquiry. One might be tempted to compare it with Ref 15, but I think Helson's manuscript goes far beyond in terms of scientific impact and clarity of results. I recommend it be published in Nature.

A few helpful suggestions:

- 1) Figure 2a needs an x-axis label that shows time as well as the pump strength from panel b.
- 2) The line denoting the phase boundary in figure 2a that the authors refer to repeatedly in the main text is unclear. Perhaps add a dashed line to show where this is and to help convince the reader that the dependence is actually linear.

Congratulations on a terrific result, exposition, and theoretical treatment.

Referee #3 (Remarks to the Author):

In this paper, Helson et al report on density-wave ordering in a Fermi gas with van der Waals type contact interaction between atoms as well as a cavity-mediated long-range interaction. The novelty of this work has two aspects: (i) together with ref. 15 it is one of the first experiments with fermions in cavities, and (ii) the detection of the density-wave via the (dynamical) susceptibility. I think this is an interesting and timely piece of work, that is by itself not too spectacular, but paves the way towards more interesting future experiments.

In these ultracold atoms, as is well known, the contact interaction alone leads to pairing (superfluid), which can be tuned across the BCS-BEC crossover via magnetic field. Here, the effect of pairing on the density-wave transition (which for the cavity system is equivalent to superradiance) is investigated, see Fig. 2c. This is clearly a novelty and a robust/valid result.

In my view, this paper is an intermediate step towards the more exciting aspect of the interplay between density-wave ordering and pairing. It will be interesting to see what happens, also from the point of view of photoemission spectroscopy with momentum and energy resolution. This will be a new step that would also go beyond what has been done in „cavity quantum materials“, for which the present work is discussed as a „cold atom proxy“. Since this seems to be one of the main motivations of the present work that would raise interest beyond the quantum-gas community, I would like to see a somewhat more comprehensive discussion of which aspects of the present and envisioned future studies with this setup could be transferred also to materials, and what are the key differences. See for the brief review <https://aip.scitation.org/doi/10.1063/5.0083825> and the perspective <https://www.nature.com/articles/s41586-022-04726-w> for recent overviews of these topics.

Author Rebuttals to Initial Comments:

Thanks for your consideration of our submitted manuscript. We also thank the three referees for their comments and positive appreciations.

As suggested by the referees, we made two minor changes in the text of the manuscript:

- Figure 2a now has been improved following the recommendations of referee 2. It now features additional timing axis and indications on the threshold.
- We have added a short (<100 words) paragraph in the discussion section (second last paragraph in the resubmitted manuscript), comparing the light-matter coupling we use with different proposals in the condensed matter physics context. As suggested by the referee, we also added three references in this discussion.